# Neutrophil Extracellular Traps Mediate Bovine Endometrial Epithelial Cell Pyroptosis in Dairy Cows with Endometritis

**DOI:** 10.3390/ijms232214013

**Published:** 2022-11-13

**Authors:** Wenxiang Shen, Xiaoyu Ma, Dan Shao, Xiaohu Wu, Shengyi Wang, Juanshan Zheng, Yanan Lv, Xuezhi Ding, Baohua Ma, Zuoting Yan

**Affiliations:** 1Lanzhou Institute of Husbandry and Pharmaceutical Science, Chinese Academy of Agricultural Science, Lanzhou 730050, China; 2College of Veterinary Medicine, Northwest A&F University, Xianyang 712000, China; 3College of Veterinary Medicine, Inner Mongolia Agricultural University, Hohhot 010018, China

**Keywords:** NETs, pyroptosis, dairy cow, endometritis, BEECs

## Abstract

Neutrophils are involved in the development of endometritis, but it remains unknown how neutrophils induce inflammation and tissue damage. Neutrophil extracellular traps (NETs) clear invading pathogens during infection but induce pyroptosis, leading to inflammation and tissue damage. Thus, our objective was to investigate whether NETs participate in bovine endometrial epithelial cell (BEEC) pyroptosis during endometritis. To confirm this, NETs and caspase-1/4; apoptosis-associated speck-like protein containing a caspase-recruitment domain(ASC); nod-like receptor protein-3 (NLRP3); and gasdermin D N-terminal (GSDMD-N), TNF-a, IL-1β, IL-6, and IL-18 in endometrial tissue were detected. Pathological section and vaginal discharge smears were performed to visually determine endometritis in the uterus. BEECs were stimulated with NETs to induce pyroptosis, which was treated with DNase I against pyroptosis. Caspase-1/4, ASC, NLRP3, GSDMD-N, TNF-a, IL-1β, IL-6, and IL-18 in BEECs were analyzed in endometrial tissue. The results showed that NET formation, as well as pyroptosis-related proteins and proinflammatory, cytokines were elevated in the endometrial tissue of cows with endometritis. Pathological sections and vaginal discharge smears showed increased neutrophils and plasma cells in the uterus, as well as tissue congestion. In BEECs, NETs increased the level of pyroptosis-related proteins and proinflammatory cytokines and were diminished by DNase I. In summary NETs participate BEEC pyroptosis during endometritis in dairy cows.

## 1. Introduction

Cow endometritis is the inflammation of the endometrium after delivery, which is caused by infection with invading microorganisms because the uterus is in an open state [1]. It affects reproductive performance through sustained inflammation and tissue damage in the endometrium, which delays the onset of ovarian cyclic activity after parturition, extends the luteal phase, and reduces the conception rate [2]. Generally, neutrophil infiltration, purulent vaginal discharge, and disruption of the endometrial epithelium are considered to constitute components in the pathophysiology of endometritis [3]. Invading microorganisms activate the innate immune system and recruit leukocytes (predominantly neutrophils) from blood vessels [4]. Neutrophils are found in the vaginal discharge and endometrium of cows with endometritis, and infiltration of the uterus is linked to endometritis and sustains the inflammatory state of the endometrium [5,6,7]. However, the potential mechanism by which neutrophils cause inflammation and tissue damage in cows with endometritis remains unclear.

Based on validated evidence, neutrophils first crawl into tissues infected with invading pathogens and clear them via phagocytosis and degranulation [8,9]. Recent research revealed a novel killing mechanism of neutrophils, whereby they extrude nuclear chromatin mixed with cytoplasmic components (including granule components), called neutrophil extracellular traps (NETs), which have a web-like structure [10]. The dominant components of NETs are histones, neutrophil elastase (NE), extracellular DNA, and myeloperoxidase (MPO) [11]. Many harmful substances and disease-causing microorganisms can cause NET formation, such as phorbol 12-myristate 13-acetate (PMA), lipopolysaccharides, bacteria, parasites, and viruses [12]. DNase I was utilized to break down NETs in a study that validated the effect of NETs on disease [13]. NETs exhibit powerful lethality as a neutrophil-killing mechanism, not only killing pathogens but also causing collateral tissue damage and aggravating inflammation [14]. 

NET-derived HMGB1 activates caspase-1-dependent macrophage pyroptosis during sepsis [15]. NETs mediate monocyte inflammatory activation and IL-1β secretion, which injures the airway epithelial cells [16]. In diabetic nephropathy, NETs promote NLR family pyrin domain-containing 3 (NLRP3) inflammasome activation and glomerular endothelial dysfunction [17]. NETs induce pyroptosis, apoptosis, and necrosis in bovine mammary epithelial cells (BMECs), resulting in impaired BMECs during bovine mastitis [18]. Thus, we hypothesized that NETs may induce pyroptosis in epithelial cells (BEECs), aggravating inflammation and tissue damage during endometritis.

Pyroptosis is a lytic programmed cell death process characterized by cell swelling and intact nuclei [19,20]. It has been reported that caspase-1 or caspase-11/4/5 is activated during pyroptosis, gasdermin D (GSDMD) is cleaved, and the N-terminal domain can oligomerize to form holes in the cell membrane and induce cell membrane rupture [21]. In the caspase-1-dependent pyroptosis pathway, pathogen-associated molecular patterns (PAMPs) or danger-associated molecular patterns (DAMPs) activate pattern recognition receptors (PRRs), leading to inflammasome activation, which recruits the inflammasome connexin, and ASC forms macromolecular complexes with the protease caspase-1. Inflammasome-activated caspase-1 directly cleaves GSDMD to initiate pyroptosis, in addition to cleaving pro-IL-1β and pro-IL-18 precursors to convert IL-1β and IL-18 into mature proinflammatory proteins [22]. In the caspase-1-independent pyroptotic pathway, LPS-activated human caspase-4 and caspase-5 or murine caspase-11 can directly cleave GSDMD and induce pyroptosis. The N-terminal fragment of GSDMD cleavage can also activate the NLRP3 inflammasome and caspase-1-dependent maturation of IL-1β and IL-18 [23]. The NLR family pyrin domain containing 3 (NLRP3) inflammasome, an important inflammasome in the process of pyroptosis, consists of NLRP3, an apoptosis-associated speck-like protein containing a CARD (ASC), and the precursor of caspase-1, which cleaves caspase-1 and caspase-4 to activate the downstream pyroptotic pathway [24]. 

In this study, we hypothesized that NETs mediate pyroptosis in bovine endometrial epithelial cells, aggravating inflammation and inducing tissue damage. To test our hypothesis, we identified cows with endometritis by counting the ratios of neutrophils in vaginal discharge smears and grouped these cows. Using uterine biopsy, we collected the endometria of dairy cows and detected NET formation and the expression of pyroptosis-related proteins (GSDMD-N, NLRP3, ASC, caspase-1, and caspase-4) in these biopsy endometrial tissues. BEECs are often used as bovine in vitro models to study biological characteristics, cellular and molecular mechanisms, and responses to infection or damage to the bovine uterine epithelium. We investigated whether NET formation is directly associated with pyroptosis of the endometrium, using NETs to stimulate BEECs and DNase I to decrease the effect of NETs on BEECs. To validate our hypothesis, we also measured the levels of proinflammatory cytokines (IL-1β and interleukin-6 (IL-6), IL-18, and tumor necrosis factor-α (TNF-α)), as well as those of pyroptosis-related proteins (GSDMD-N, NLRP3, ASC, caspase-1, and caspase-4), in BEECs.

## 2. Results

### 2.1. The Ratio of Neutrophils in Vaginal Discharge Smear 

To rapidly diagnose cow endometritis, vaginal discharge smears were collected at 28 to 35 d postpartum and stained with Diff-Quick. As shown in Figure 1A, cows with endometritis had significantly more neutrophils in their vaginal secretions than cows in the control group. 

### 2.2. Histopathological Examination

To observe the changes in the histopathological characteristics of cow endometrial tissue during endometritis, according to the 2.1 grouping method, we detected the biopsy endometrial tissue of different groups by H&E staining and obtained the results shown in Figure 1C. The endometrial tissues from the biopsy of cows with endometritis showed a large number of neutrophils, few plasma cell infiltrates, and minimal tissue congestion, with marked enlargement and detachment of epithelial cells. In the healthy group, the endometrium had no obvious neutrophil or plasma cell infiltration, a few red blood cells were observed, and the endometrial epithelial cells were well-shaped.

### 2.3. Proinflammatory Cytokines Increased in Endometritis Tissues

Proinflammatory cytokine detection was performed to compare the differences between the endometritis and healthy groups. IL-6 and TNF-α are representative proinflammatory cytokines, whereas IL-1β and IL-18 are mainly mature during pyroptosis. As shown in Figure 2, cow endometritis increased endometrial tissue IL-1β levels by 16.5-fold (i.e., from 2.3 to 33.1 ng/L) (Figure 2A), increased endometrial tissue IL-6 levels by 3-fold (i.e., from 64.7 to 204.9 pg/mL) (Figure 2B), increased endometrial tissue IL-18 levels by 2-fold (i.e., from 14.7 to 30.9 pg/mL) (Figure 2C), and increased endometrial tissue TNF-α levels by 4-fold (i.e., from 25.5 to 107.4 ng/L) (Figure 2D). These results suggest that proinflammatory cytokine levels increase and pyroptosis may occur in the endometrium during cow endometritis.

### 2.4. NET Formation in Endometrial Tissue during Cow Endometritis

NETs are composed of extracellular DNA, neutrophil-derived granular proteins, and histones. To detect the presence of NETs in the endometrial tissue of cows with endometritis, neutrophil elastase (NE), histone 1, and DNA were colocalized in endometrial tissue by immunofluorescence staining. The formation of web-like structures was observed in the endometritis tissue, and no web-like structures were detected in the healthy control group (Figure 3), indicating that NET formation in the endometrium may be associated with endometritis in cows.

### 2.5. Endometrial Tissue Undergoes Pyroptosis in Cows with Endometritis

To determine whether the endometrial tissues of cows with endometritis suffered from pyroptosis, we examined the key molecules involved in pyroptosis. The expressions of caspase-1, caspase-4, ASC, GSDMD-N, and NLRP3 in endometrial tissue were measured using immunohistochemistry and Western blotting. According to immunohistochemistry, the expression of caspase-1 in endometritis tissues was increased by sixfold compared to the healthy group (Figure 4B), and the expression of caspase-4 was found to increase by sevenfold compared to the healthy group (Figure 4C). The expression of ASC was increased by sixfold compared to the healthy group (Figure 4D), the expression of GSDMD-N was increased by sixfold compared to the healthy group (Figure 4E), and the expression of NLRP3 level was increased by threefold compared to the healthy group (Figure 4F). According to Western blotting, the expression of caspase-1 in endometritis tissue increased by 41.8% compared with the healthy group (Figure 5B), the expression of caspase-4 increased by 34.7% compared with the healthy group (Figure 5C), the expression of ASC was increased by 23.2% compared with the healthy group (Figure 5D), the expression of GSDMD-N was increased by 24.1% compared with the healthy group (Figure 5E), and the expression of NLRP3 was increased by 25.4% compared with the healthy group (Figure 5F).

### 2.6. NETs Induce BEEC Cytotoxicity

BEEC is a common research material used to study uterine diseases. In this study, to determine the role of NETs in the uterus, BEECs were subjected to NET stimulation for 16 h with or without DNase I to investigate whether NETs caused BEEC cytotoxicity. NETs (500 ng/mL) were coincubated with BEECs, and DNase I (25 mg/mL) was administered to break down NETs. An LDH assay revealed a significant increase in the NET-stimulated groups, and DNase I inhibited the LDH increase in the supernatant of BEECs. Under NET stimulation, the level of BEEC cytotoxicity increased by 72% relative to the control group, and administration of DNase I caused a 47.2% reduction in cytotoxicity (Figure 6E). These results suggest that NETs induced BEEC cytotoxicity.

### 2.7. Proinflammatory Effects of NETs on BEECs

To verify whether NETs are among the factors causing BEEC inflammation, we treated BEECs with NETs and NETs + DNase I to detect the production of the proinflammatory cytokines IL-1β, IL-6, IL-18, and TNF-α. Under NET stimulation, the level of BEEC supernatant IL-1β increased by fivefold relative to the control group, and the administration of DNase I caused a 40% reduction in IL-1β (Figure 6A); the level of BEEC supernatant IL-6 increased by sevenfold relative the control group, and the administration of DNase I caused a 37.5% reduction in IL-6 (Figure 6B); the level of BEEC supernatant IL-18 increased by 5.6-fold relative to the control group, and administration of DNase I caused a 33.6% reduction in IL-18 (Figure 6C); the level of BEEC supernatant TNF-α increased by ninefold relative to the control group, and the administration of DNase I caused a 25% reduction in TNF-α (Figure 6D). The results reported herein indicate that NETs induce inflammation in BEECs.

### 2.8. NETs Induced BEEC Pyroptosis through the Activation of NLRP3, ASC, Caspase-1, and Caspase-4 

To verify whether NETs induce BEEC pyroptosis, we observed antibodies against caspase-1, caspase-4, ASC, GSDMD, and NLRP3 conjugated with green, fluorescent secondary antibodies under a confocal laser microscope. 

Immunofluorescence revealed that NET increased the expression of caspase-1, caspase-4, ASC, GSDMD-N, and NLRP3 in BEECs by 92.5%, 68.9%, 75.6%, 91.1%, and 85.6%, respectively, compared with the control group, and administration of DNase I caused a 70.2%, 40.9%, 39.5%, 37.2%, and 53.3% reduction in caspase-1, caspase-4, ASC, GSDMD, and NLRP3, respectively (Figure 7B–F).

Western blotting revealed that NET increased the expression of caspase-1 in BEECs by 54.3% compared with the control group, and the administration of DNase I caused a 36.2% reduction in caspase-1 (Figure 8B); NET increased the expression of caspase-4 in BEECs by 57.1% compared with the control group, and the administration of DNase I caused a 34.5% reduction in caspase-4 (Figure 8C); NET increased the expression of ASC in BEECs by 53.6% compared with the control group, and the administration of DNase I caused a 28.9% reduction in ASC (Figure 8D); NET increased the expression of NLRP3 in BEECs by 58.8% compared with the control group, and the administration of DNase I caused a 47.1% reduction in NLRP3 (Figure 8E); NET increased the expression of GSDMD-N in BEECs by 36.2% compared with the control group, and the administration of DNase I caused a 43.3% reduction in GSDMD-N (Figure 8F). Furthermore, NET-stimulated BEECs were enlarged with PI-stained nuclei and morphologically consistent with features of pyroptosis by PI and Annexin V staining (Appendix A). Overall, these results indicate that NETs participate in BEEC pyroptosis.

## 3. Discussion

The aim of this study was to determine whether NETs induce pyroptosis and cause tissue damage and inflammation in BEECs during cow endometritis. Endometrial biopsy of dairy cows revealed that the endometrial tissue of dairy cows with endometritis had inflammatory cell infiltration with epithelial cell exfoliation compared with healthy dairy cows, and NET formation occurred only in endometritis tissue. Qualitative and quantitative analysis of major pyroptosis-related proteins (caspase-1, caspase-4, ASC, GSDMD-N, and NLRP3) in endometrial tissues showed that pyroptosis occurred in endometrial tissues of cows with endometritis. An in vitro study was conducted with BEECs stimulated with NETs to verify whether NETs were the cause of endometrial pyroptosis. The results showed that NET-induced BEECs underwent pyroptosis, resulting in cell damage and increased inflammatory cytokines. NET disruption by DNase I alleviated NET-induced pyroptosis and reduced cell damage and inflammation, confirming that NETs may induce and aggravate tissue damage and inflammation during endometritis. Disruption of NETs represents a possible strategy for the treatment of endometritis. 

Neutrophils are the dominant component of leukocytes in the innate immune system and affect endometrial function in dairy cattle. Neutrophil infiltration is one of the main causes of endometritis, and the persistent presence of neutrophils causes local inflammation of the uterus [25]. The ratio of neutrophils in vaginal discharge is a classical diagnostic method for endometritis in dairy cows [26,27]. Neutrophils, through the release of neutrophil extracellular traps (NETs), clear invading pathogens and are an important effect mechanism [28]. NET formation in the uterus was observed during mare endometritis [29]. In this study, NETs were detected in the endometrial biopsy tissue of cows with endometritis but not in the endometria of healthy cows. This is evidence of NET formation in the endometria of cows with endometritis. Recently, we found that NETs limit the spread of invading pathogens and kill them using the antibacterial proteins with which they are decorated [28]. However, the excessive formation of NETs causes tissue damage and inflammation in some diseases, such as acute pancreatitis [30] and inflammatory bowel disease [31]. We also found that NETs are involved in the pathological process of endometriosis in women [32] and endometritis in mice [13]. Therefore, we investigated whether NETs cause endometrial tissue damage and inflammation in dairy cows. First, we examined the histological changes in the endometrial tissue of cows from a biopsy, which showed inflammatory cell infiltrates and disorders of the endometrial epithelial cells during endometritis. The formation of NETs in the endometrial tissue of cows with endometritis was observed by immunofluorescence. However, this evidence is insufficient to demonstrate a causal relationship between NETs and inflammation, as well as pathological damage to endometrial tissue in cows. Thus, we isolated neutrophils from the peripheral blood of cows and induced neutrophils to form NETs by PMA (phorbol 12-myristate 13-acetate, PMA), and the isolated NETs were coincubated with BEECs. Using an LDH assay to investigate the cytotoxicity of NETs, we established NET cytotoxicity on BEECs and inhibited NETs, which was significantly increased in the supernatant of BEECs stimulated by NETs and decreased by DNase I, implying that NETs can induce BEEC cytotoxicity. In addition, NETs increased the levels proinflammatory cytokines in BEEC supernatant. These pieces of evidence imply that NETs are involved in tissue damage and inflammation of the endometrial epithelium in cows. However, the underlying mechanism by which NETs cause endometrial tissue damage and inflammation is still unknown.

Pyroptosis is programmed cell death via cell lysis, which can cause severe inflammation and tissue damage [33]. NETs are a mixture, the main components of which, i.e., extracellular DNA [16], histones [34], HMGB1 [15], neutrophil elastase [35], etc., are released extracellularly as DAMPs to induce pyroptosis and cause severe inflammation and tissue damage. Previous studies have demonstrated that NETs can induce macrophage pyroptosis, elevate caspase-1 with increasing NETs dose, and increase inflammatory cytokines IL-1β and TNF-α in sepsis [15]. In acute respiratory distress syndrome, NETs aggravate alveolar macrophage pyroptosis, and ASC foci, caspase-1, IL-1β, IL-6, and TNF-α are significantly associated with elevated NETs [36]. Paul Kelly et al. reported that endometritis was accompanied by elevated IL-1β and used experiments to prove that NLRP3 and caspase-4 play a major role in the expression of IL-1β. Interestingly, caspase-1 does not play a role in this process [37]. It has been reported that corneal epithelial cell damage can be effectively alleviated by inhibiting the NLRP3-ASC-caspase-1-GSDMD pyroptosis pathway in dry eye disease [38]. In addition, during NET formation in vivo, because its main component is DAMPs, it can be recognized by patterns on the surface of endometrial epithelial cells and then activate the assembly of the intracellular NLRP3 inflammasome, which consists of NLRP3, ASC, and procaspase 1 [39]. Activated NLRP3 inflammasome cleaves activated procaspase 1 into mature caspase 1, which cleaves pro-IL-1β and pro-IL-18, as well as porogen precursor GSDMD. The cleaved IL-1β and IL-18 can be released extracellularly to aggravate inflammation and cause tissue damage. GSDMD is cleaved into N-segment polypeptide GSDMD-N with a pore-forming effect, which leads to the outflow of cell contents and cell rupture, resulting in a local or systemic inflammatory response [33]. HMGB1 in NETs can assist caspase 4 activation to induce pyroptosis [40]. We speculate that caspase 4 may be activated by NETs during endometritis and cleave pro-IL-1β, pro-IL-18, and GSDMD to induce pyroptosis. Therefore, we measured caspase-1/4, ASC, NLRP3, and GSDMD-N levels to determine whether pyroptosis occurs in endometritis. These results suggest that pyroptosis occurs in the endometrium during endometritis. To investigate the direct effect of NETs on the endometrium, we used BEECs isolated from bovine uteri coincubated with NETs and NETs deleted by DNase I. We obtained the same results in BEECs stimulated with NETs compared to those in endometrial tissue during endometritis; the levels of pyroptosis-related proteins were increased. 

NET-derived materials, such as high-mobility group box 1 (HMGB1) [41], histone [42], extracellular DNA [34], myeloperoxidase (MPO) [43], and neutrophil elastase (NE) [44], are classical DAMPs that mediate tissue damage and hyperinflammatory responses in inflammatory diseases. In this study, we used NETs from peripheral cow blood induced by PMA, which is not a single component of NETs and is more convincing proof of NET function (Appendix A). NETs, as DAMP molecules, not only cause pyroptosis but also mediate other types of cell death. Here, we found that NETs mediated BEEC pyroptosis during endometritis and that NETs are a possible therapeutic target for endometritis. Unfortunately, because the large number of components of NETs that can induce pyroptosis could not be included in a single study, in the present study, we only investigated the effect of NET components on endometrial epithelial cells as a whole. However, the results of this study lay the foundation for subsequent studies on the effect of NETs on endometrial epithelial pyroptosis, and future studies can be conducted to investigate the mechanism of endometrial epithelial scorching by specific NET components.

## 4. Materials and Methods

### 4.1. Animals and Grouping

This study included 14 Holstein cows from a commercial dairy herd in Lanzhou, Gansu Province, China. All cows included in this study were 3–6 years old (2–5 parities), without any systemic clinical signs, and they were sampled 28–35 days postpartum. The neutrophil ratio in a vaginal discharge smear (≥18% indicates endometritis in cows) is the primary basis for assessing the presence of endometritis in cows. The cows were divided into two groups. The neutrophil ratios were < 18% in the healthy cow group and ≥18% in the endometritis group. The key materials used in this test can be seen in Table 1.

This study was approved by the Institutional Animal Care and Use Committee of the Lanzhou Institute of Husbandry and Pharmaceutical Sciences at the Chinese Academy of Agricultural Sciences (syxk-2020-0002).

### 4.2. Vaginal Discharge Smear

Vaginal discharge smears were prepared immediately after the collection of vaginal discharge samples. The slides were air-dried and stained using the Diff-Quick staining protocol (Solarbio, Beijing, China). The smears were evaluated under a microscope, and the proportion of neutrophils was recorded. All cells were counted, and the percentage of neutrophils was calculated based on an average of 200 cells.

### 4.3. Endometrial Tissue Collection

Uterine tissue was obtained from cows via biopsy. Briefly, lifting the tail and cleaning the vulva, cow uterine sampling forceps were guided through the rectum into the uterine horn by a veterinarian to obtain a biopsy sample. Endometrial tissues obtained through biopsy were divided into two parts: one was kept in liquid nitrogen for total protein extraction, and the other was fixed in 4% paraformaldehyde (PFA) solution for hematoxylin and eosin (HE) staining, immunohistochemistry, and immunofluorescence.

### 4.4. HE Staining

Endometrial tissues fixed with 4% PFA were embedded in paraffin and cut into 6 um thick sections. After H&E staining, pathological changes in the uterus were observed under an Olympus BX43 microscope and evaluated according to histological features (including disruption of the surface epithelium, infiltration with inflammatory cells, and vascular congestion).

### 4.5. Immunohistochemistry

Paraffin-embedded endometrial tissue was cut into 6 um thick slices, dewaxed twice in xylene for 20 min each time, and rehydrated. The sections were then treated with 3% H2O2 for 30 min to inactivate endogenous peroxidase and washed with PBS. The sections were placed in a pressure cooker containing Tris-EDTA (pH 9.0) and boiled for 2 min, after which the heat source was removed, and the sections were left at room temperature for 15 min. The lid of the pressure cooker was then opened and cooled to room temperature. The sections were blocked with a blocking buffer for 30 min at a temperature of 37 °C. Next, the sections were incubated with primary antibody at 4 °C overnight. The following day, the sections were incubated with goat anti-rabbit IgG H&L (HRP) (1:200, Abcam, Cambridge, UK) for 30 min at 37 °C. After washing thrice with PBS, the sections were stained using a DAB chromogenic kit (ZSGB-BIO, Beijing, China) for 5 min. The sections were then rinsed with running water, stained with hematoxylin (Solarbio, Beijing, China) for 6 min, fractionated with 1% hydrochloric acid for 30 s, dehydrated with gradient alcohol, made transparent with xylene, and sealed with a water-soluble sealer (ZSGB-BIO, Beijing, China). The positive area of the related proteins was measured by Image J software, and three images were used for statistical analysis.

### 4.6. BEEC Isolation

In our previous study, bovine endometrial epithelial cells (BEECs) were isolated from the healthy uterus of a 6-month-old dairy cow [3]. Briefly, a healthy uterus was placed in the laboratory in sterile PBS (pH 7.2) containing 1× antibiotic-antimycotic (100×). The endometrial tissue near the uterine horns was collected and washed twice in PBS (pH 7.2) and treated with 1% collagenase I (Sigma-Aldrich, Chicago, IL, USA) diluted in DMEM/F12 for 6 h. The digested endometrium was scraped using a sterile cell scraper, and the scraped material was collected and washed with PBS (pH 7.2). The collected material was centrifuged at 100× *g* for 5 min to collect the cell suspension. Trypan blue staining was used to assess the cell viability. The cells were cultured in DMEM/F12 containing 10% fetal bovine serum and a 1× antibiotic–antimycotic. Then, they were incubated in a 37 °C, 5% CO2 incubator with medium changes every three days until the cells reached approximately 90% confluence. BEEC purification was assessed by the expression of epithelial-specific rabbit anti-cytokeratin-18 monoclonal antibody (1:2000, Proteintech, Chicago, IL, USA) and goat anti-rabbit IgG H&L (1:500, Abcam, Cambridge, UK) fluorescent secondary antibody (Appendix A).

### 4.7. Neutrophil Isolation and Induction of NET Formation 

To isolate neutrophils from peripheral blood, we collected 50 mL of venous blood from the tail vein of the cow in an anticoagulant tube containing sodium heparin. The specific neutrophil isolation process is shown in Appendix A. Isolated neutrophils were resuspended and cultured in RPMI 1640 medium (Gibco, Grand Island, NY, USA).

Isolated dairy cow neutrophils were seeded at a density of 2 × 10^5^ cells/well in 6-well plates. After incubation for 30 min at 37 °C, the neutrophils were incubated with 12-phorbol 13-myristate acetates (PMA) (50 nmol/L, Sigma-Aldrich, Chicago, IL, USA) for 4 h to form NETs. The neutrophils were resuspended and centrifuged for 5 min (50× *g*), and the supernatant was assayed for NET concentration using a Quant-iT™ PicoGreen kit (Invitrogen, Carlsbad, CA, USA). In addition, we observed NET formation in PMA-stimulated neutrophils by immunofluorescence (Appendix A).

### 4.8. BEECs Stimulated with NETs and Cytotoxicity Assay 

BEECs were seeded at a density of 2 × 10^5^ cells/well in 6-well plates and incubated for 24 h. Then, the BEECs were treated for 16 h with the following treatments: control (equivalent medium), NET (500 ng/mL), NET (500 ng/mL) + DNase I (25 µg/mL), and DNase I (25 µg/mL, Solarbio, Beijing, China). Cell viability was assessed using a CyQUANT LDH cytotoxicity detection kit (Invitrogen, Carlsbad, CA, USA). The cell supernatant was collected, and the inflammatory factor content was determined by ELISA. The expression and translation levels of pyroptosis-related proteins (caspase-1, caspase-4, ASC, GSDMD-N, and NLRP3) were detected using real-time quantitative PCR and Western blotting, respectively.

### 4.9. Immunofluorescence

For the detection of NETs in the endometrium, paraffin-coated endometrial tissue was cut into 6 μm sections, dewaxed, and rehydrated. After deparaffinization in xylene, the sections were rehydrated, and antigen retrieval was performed by a blocking in blocking buffer for 1 h. The sections were incubated with mixed primary antibodies for double labeling: rabbit anti-neutrophil elastase polyclonal antibody (1:200, Abcam, Cambridge, UK) and mouse anti-DNA-Histone1 monoclonal antibody (1:200, Millipore, MA, USA) overnight at 4 °C. The sections were then incubated with mixed fluorescently conjugated secondary antibodies (1:200, Alexa Fluor 594-conjugated goat anti-mouse IgG H&L antibody and Alexa Fluor^®®^ 488 goat anti-rabbit IgG H&L) for one hour at room temperature and stained with DAPI (Solarbio, Beijing, China) for 5 min. The slides were mounted with an antifluorescence quencher and observed under a Zeiss LSM 800 confocal laser scanning microscope (Oberkochen, Germany). 

Isolated neutrophils were seeded at a density of 2 × 105 cells/well in 24-well plates. After incubation for 2 h at 37 °C, 50 nmol/L PMA (Sigma-Aldrich, Chicago, IL, USA) was added, and incubation was continued for 4 h to form NETs. The cells were fixed with 4% paraformaldehyde and blocked with blocking solution at room temperature for 1 h; then, the primary antibody was incubated overnight at 4 °C. The next day, fluorescent secondary antibody was added for 1 h, DAPI was added at room temperature for 5 min, and an antifluorescence quencher was added for confocal observation.

Isolated BEECs were seeded at a density of 2 × 105 cells/well in 24-well plates and placed in a 5% carbon dioxide incubator at 37 °C overnight. According to different groupings, blank medium, NET (500 ng/mL), NET (500 ng/mL) + DNase I (25 μg/mL), and DNase I (25 μg/mL, Solarbio, Beijing, China) were added to treat the BEECs (all treatment drugs were diluted with medium). The BEECs were cultured for 16 h and fixed with 4% paraformaldehyde. To detect the effects of NETs on BEEC morphology, we stained the BEECs using an annexin V-FITC/PI kit (Roche, Basel, Switzerland). To detect whether NETs caused BEEC pyroptosis, we used primary antibodies of rabbit anti-caspase-1 antibody (1:200, Proteintech, Chicago, IL, USA), rabbit anti-caspase-4 antibody (1:200, Affinity bioscience, Liyang, Jiangsu, China), rabbit anti-ASC antibody (1:200, Proteintech, Chicago, IL, USA), anti-NLRP3 antibody (1:200, Proteintech, Chicago, IL, USA), and rabbit anti-GSDMD antibody (1:200, Affinity bioscience, Jiangsu, China) with each group. The BEECs were incubated overnight at 4 °C and incubated with Alexa Fluor^®®^ 488 goat anti-rabbit IgG H&L (Abcam, Chicago, IL, USA) for 1 h at 37 °C. After the nuclei were stained with DAPI, confocal images were obtained to calculate the fluorescence intensity of each group. Three images were selected for the measurement of fluorescence intensity using Olympus FV10-ASW software (Olympus).

### 4.10. Enzyme-Linked Immunosorbent Assay (ELISA)

The levels of interleukin-6 (IL-6), interleukin-1β (IL-1β), interleukin-18 (IL-18), and tumor necrosis factor-α (TNF-α) (Jinma Co., Ltd., Shanghai, China) were measured in the supernatant of uterine tissue homogenates and the supernatant of the BEECs (stimulated by NETs) using ELISA kits (Jinma, Shanghai, China) according to the manufacturer’s instructions. Absorbance was measured at 450 nm using a Multiskan™ GO full-wavelength microplate reader (Thermo Fisher Scientific, Rockford, IL, USA). First, the uterine tissue, stored at −80 °C, was weighed at 0.1 g in a 2 mL tube, mixed with 500 µL of RIPA lysis buffer (Solarbio, Beijing, China) containing 1% protease inhibitor cocktail I (Medchem Express, Monmouth Junction, New Jersey, USA), and ground with a tissue homogenizer. Grinding was performed until no intact tissue fragments were visible; all steps of the grinding process were performed on ice. The supernatant was collected, centrifuged, and stored in a −20 °C refrigerator (15,000× *g*, 4 °C, and 10 min).

### 4.11. Western Blotting

The total protein was extracted from bovine endometrial tissues and BEECs using a high-performance RIPA lysis solution (tissue/cell) (Solarbio, Beijing, China). A BCA protein assay kit (Takara, Tokyo, Japan) was used to determine the protein concentration. Based on the BCA assay results, the protein contents in each group were adjusted to the same concentration using sterile water. The total proteins were denatured in a metal bath at 100 °C with a 5× loading buffer (Solarbio, Beijing, China) for 15 min. The total proteins (30 µg) were separated by FuturePAGETM 4–20% precast gel (ACE Biotechnology, Nanjing, China) and transferred to a 0.45 um nitrocellulose membrane (Solarbio, Beijing, China). The membranes were blocked with QuickBlock Western Blocker (Beyotime Biotechnology, Shanghai, China) for 15 min at room temperature and incubated with primary antibody at 4 °C overnight. The next day, the primary antibody was recovered and incubated with fluorescent secondary antibodies IRDye 800CW goat anti-rabbit IgG (H + L) (1:10,000, Gene Company Limited, Hong Kong, China) and IRDye 680RD goat anti-mouse IgG (H + L) (1:10,000, Gene Company Limited, Hong Kong, China) for 1 h at 37 °C. Protein bands were observed using an LI-COR Odyssey CLx near-infrared two-color laser imaging system (Gene Company Limited, Hong Kong, China). The following primary antibodies were used: anti-β-actin (1:2000, Abcam, Cambridge, UK), rabbit anti-caspase-1antibody (1:500, Proteintech, Chicago, IL, USA), rabbit anti-caspase-4 (1:500, Affinity Bioscience, Liyang, Jiangsu, China), rabbit anti-ASC (1:500, Proteintech), anti-NLRP3 (1:500, Proteintech), and rabbit anti-GSDMD (1:500, Affinity Bioscience).

### 4.12. Statistical Analysis

The data are expressed as the mean ± SD and were analyzed using GraphPad Prism 7 software (GraphPad Prism, San Diego, CA, USA). The unpaired Student’s t-test was used to compare the results between the healthy group and the endometritis group, as well as between differently treated cells. Western blotting and immunohistochemistry results were analyzed using ImageJ software (National Institutes of Health, Bethesda, MD, USA). Statistical significance was set at *p* < 0.05, and *n* represents the number of animals.

## 5. Conclusions

In conclusion, we identified the formation of NETs during cow endometritis and found that NETs were involved in BEEC pyroptosis and enhanced inflammation. Moreover, disruption of NETs with DNase I can effectively alleviate NET-induced BEEC damage and inflammation, suggesting that NETs as a possible target for the treatment of endometritis.

## Figures and Tables

**Figure 1 ijms-23-14013-f001:**
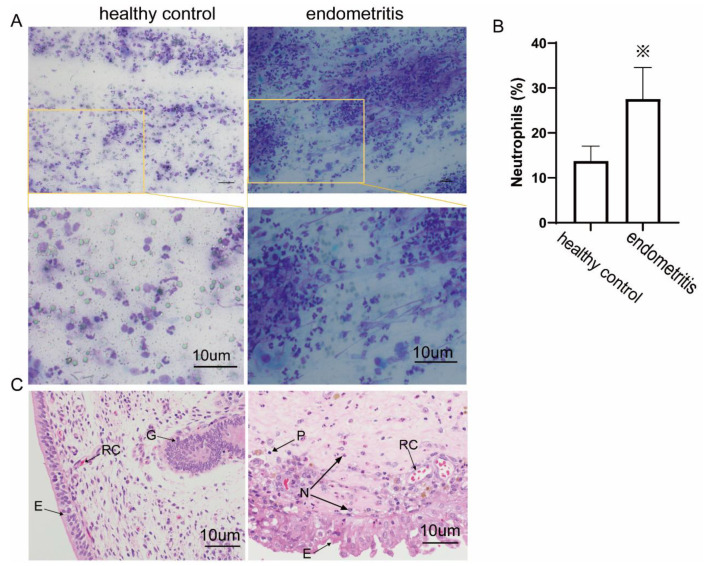
Endometritis-induced neutrophil recruitment in vaginal discharge and endometrial tissue. (**A**) Diff-Quick staining images of neutrophils in vaginal discharge; lobulated nucleus-shaped cells are neutrophils; black bar = 10 µm. (**B**) Ratio of neutrophils in vaginal discharge; endometritis group, *n* = 10; healthy group, *n* = 4. ※ *p* < 0.05. (**C**) Histopathological characteristic images of endometrial biopsy specimens from postpartum cows. E, endometrial epithelium; G, uterine glands; N, neutrophils (lobulated nucleus-shaped cells); RC, red cells; P, plasma cells (deeply stained nuclei, wheel-mounted, with less cytoplasm); 400× magnification.

**Figure 2 ijms-23-14013-f002:**
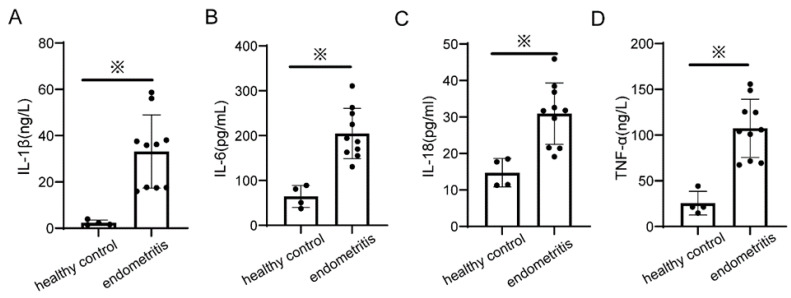
ELISA results from healthy control (*n* = 4) endometrial tissue and endometritis group (*n* = 10) endometrial tissue, showing levels of (**A**) IL-1β, (**B**) IL-6, (**C**) IL-18, and (**D**) TNF-α. Values are presented as mean ± SD. ※ *p* < 0.05. Black points represent individual data points.

**Figure 3 ijms-23-14013-f003:**
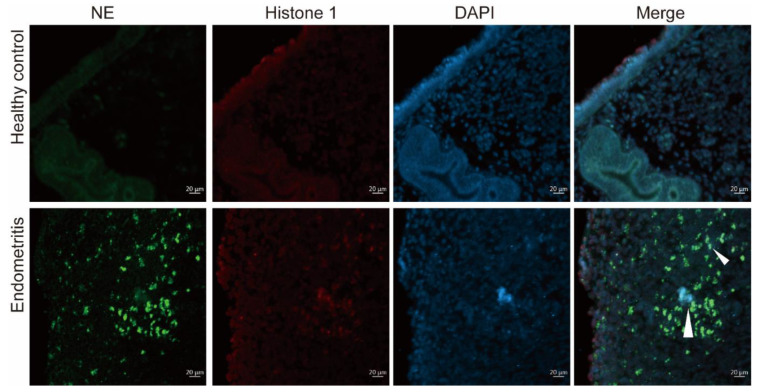
Immunofluorescent staining of NETs was assessed by confocal microscopy in healthy control endometrial tissue and endometritis group endometrial tissue, NE (green), neutrophil elastase; histone 1 (red); DAPI (blue); white arrows indicate neutrophil extracellular traps; white bar = 20 µm; 400× magnification.

**Figure 4 ijms-23-14013-f004:**
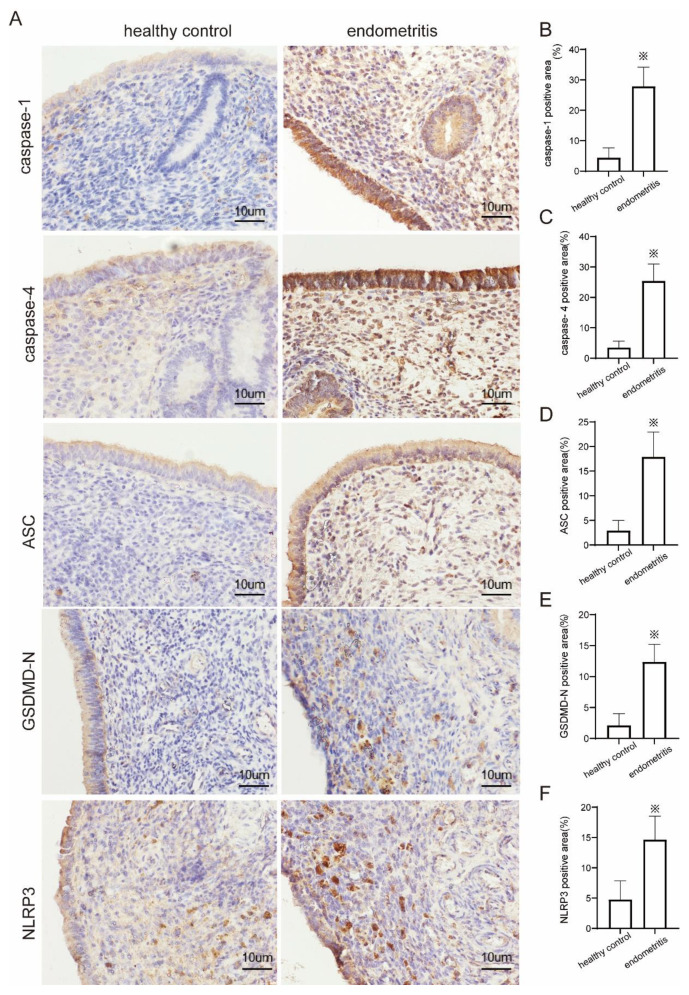
Immunohistochemical staining results of pyroptosis-related proteins in cow uterus. (**A**) Representative immunohistochemistry staining for caspase-1, caspase-4, ASC, GSDMD-N, and NLRP3 in cow uterine tissue classified as the healthy group (*n* = 3) and the endometritis group (*n* = 3); 400× magnification. (**B**) The expression of caspase-1 in the treatment groups. (**C**) The expression of caspase-4 in the treatment groups. (**D**) The expression of ASC in the treatment groups. (**E**) The expression of GSDMD-N in the treatment groups. (**F**) The expression of NLRP3 in the treatment groups. The figures were analyzed using Image J. Values are presented as the mean ± SD. The experiment was conducted with the endometrial tissues of three healthy cows and the endometrial tissues of three cows with endometritis. ※ *p* < 0.05.

**Figure 5 ijms-23-14013-f005:**
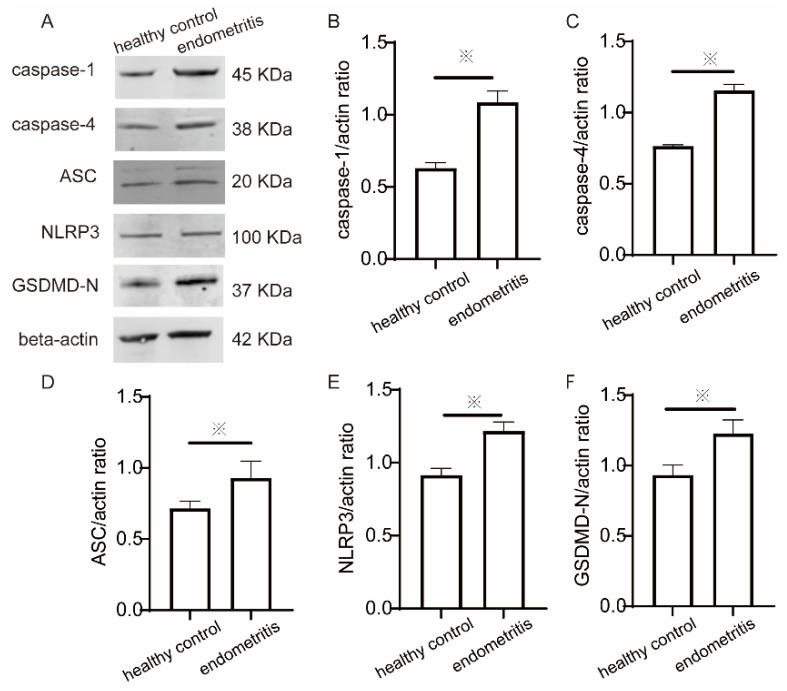
Expression of caspase-1, caspase-4, ASC, GSDMD-N, and NLRP3 in endometrial tissues was detected using Western blot. (**A**) The expression blots of pyroptosis-related proteins, including caspase-1, caspase-4, ASC, GSDMD-N, and NLRP3, with varying molecular weights. Western blot analysis of (**B**) caspase-1, (**C**) caspase-4, (**D**) ASC, (**E**) NLRP3, and (**F**) GSDMD-N in the treatment groups. The experiment was conduced with the endometrial tissues of three healthy cows and the endometrial tissues of three cows with endometritis. ※ *p* < 0.05.

**Figure 6 ijms-23-14013-f006:**
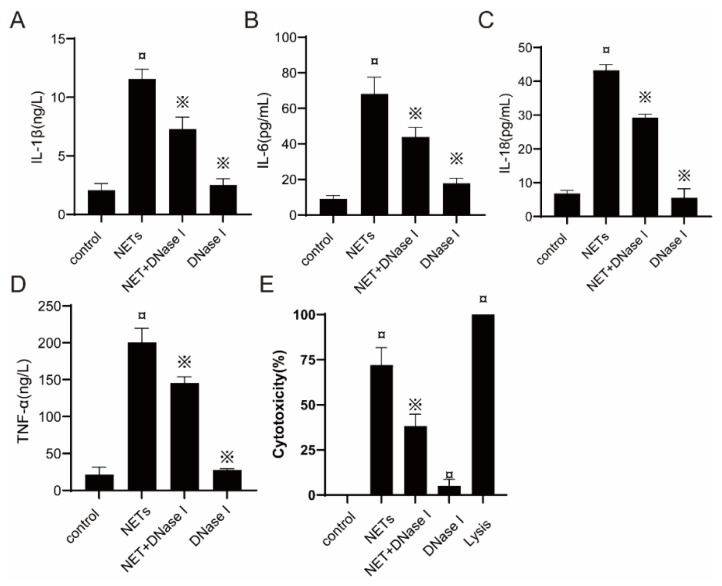
Cytokine production and cytotoxicity of untreated bovine endometrial epithelial cells (BEECs) (control), NET-treated BEECs (NET), NET- and DNase I-treated BEECs (NET+ DNase I), and DNase I-treated BEECs (DNase I). The BEECs were incubated with NETs or DNase I for 16 h. The levels of (**A**) IL-1β, (**B**) IL-6, (**C**) IL-18, and (**D**) TNF-α in BEEC supernatants were detected by ELISA. (**E**) The cytotoxicity of BEECs was determined after treatment with NET (500 ng/mL), NET (500 ng/mL) + DNase I (25 µg/mL), or DNase I (25 µg/mL) for 16 h. Lysis provided by LDH cytotoxicity assay kits was used as a positive control. Values are presented as the mean ± SD (*n* = 3). ¤ *p* < 0.05 vs. control group and ※ *p* < 0.05 vs. NETs without DNase I.

**Figure 7 ijms-23-14013-f007:**
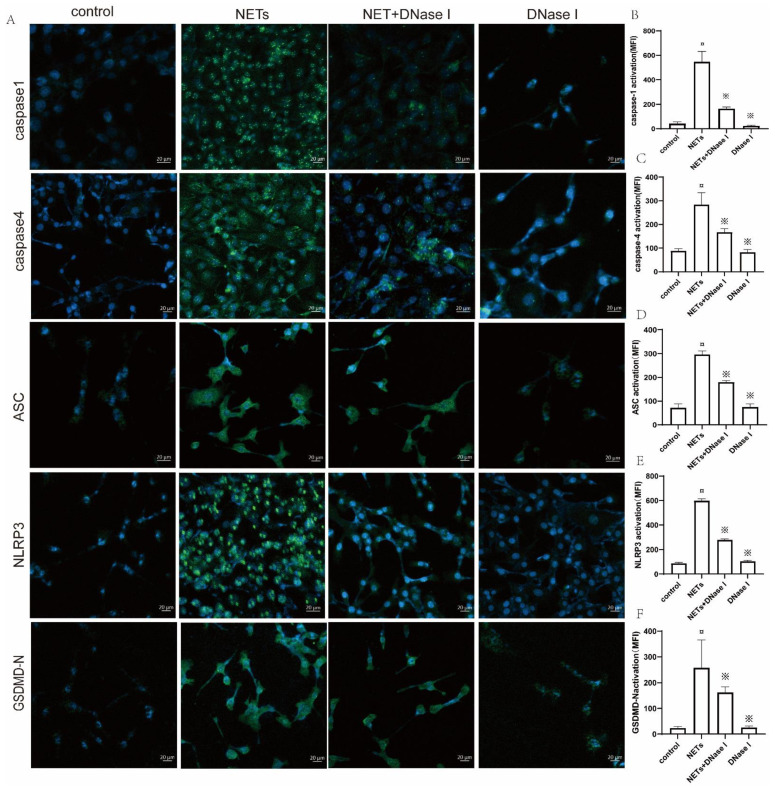
Representative immunofluorescent staining images for caspase-1, caspase-4, ASC, NLRP3, and GSDMD-N in untreated BEECs (control), NET-treated BEECs (NET), NET- and DNase I-treated BEECs (NET+ DNase I), and DNase I-treated BEECs (DNase I). (**A**) BEECs were subjected to immunofluorescent staining after treatment with NET (500 ng/mL), NET (500 ng/mL) + DNase I (25 µg/mL), or DNase I (25 µg/mL) for 16 h. The nuclei of BEECs were stained with DAPI (blue) and primary antibody (caspase-1, caspase-4, ASC, NLRP3, and GSDMD-N) conjugated with secondary antibody Alexa Fluor^®®^ 488 (green). (**B**–**F**) The mean fluorescence intensity (MFI) of activated (**B**) caspase-1, (**C**) caspase-4, (**D**) ASC, (**E**) NLRP3, and (**F**) GSDMD-N was detected and analyzed using confocal microscopy and an OLYMPUS Fluoview Ver.1.7c system. Values are presented as the mean ± SD (*n* = 3). ¤ *p* < 0.05 vs. control group and ※ *p* < 0.05 vs. NETs without DNase I; white bar = 20 µm.

**Figure 8 ijms-23-14013-f008:**
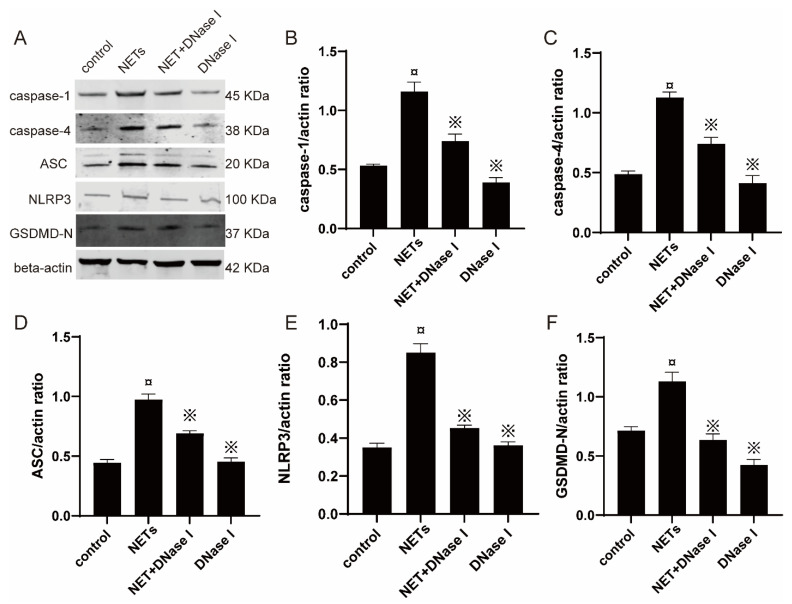
Expression of caspase-1, caspase-4, ASC, GSDMD-N, and NLRP3 in BEECs was detected by Western blotting. (**A**) The expression blots of pyroptosis-related proteins, including caspase-1, caspase-4, ASC, GSDMD, and NLRP3, with varying molecular weights. Western blot analysis of (**B**) caspase-1, (**C**) caspase-4, (**D**) ASC, (**E**) NLRP3, and (**F**) GSDMD-N in the treatment groups (*n* = 3). The gray values in the figure were analyzed using Image J. Values are presented as the mean ± SD. ¤ *p* < 0.05 vs. control group and ※ *p* < 0.05 vs. NET without DNase I.

**Table 1 ijms-23-14013-t001:** Key materials.

Reagents or Antibodies	Source	Identifier
DAB Chromogenic Kit	Zsgb-Bio, Beijing, China	Zli-9017
Water-Soluble Sealer	Zsgb-Bio, Beijing, China	Zli-9551
Cyquant LDH Cytotoxicity Assay Kit	Invitrogen, Carlsbad, CA, USA	C20300
QuickBlock^TM^ Western Blocker	Beyotime Biotechnology, Shanghai, China	P0252
5× Loading Buffer	Beyotime Biotechnology, Shanghai, China	P0015
Il-1β Elisa Kit	Jinma Co., Ltd., Shanghai, China	Jb207-Bv
Il-6 Elisa Kit	Jinma Co., Ltd., Shanghai, China	Jb204-Bv
Il-18 Elisa Kit	Jinma Co., Ltd., Shanghai, China	Jb307-Bv
TNF-α Elisa Kit	Jinma Co., Ltd., Shanghai, China	Jb010-Bv
Annexin V-FITC/PI Kit	Roche, Basel, Swiss	11858777001
FuturePAGE^TM^ 4–20% Precast-Gel	ACE Biotechnology, Nanjing, China	F11420lgel
Diff-Quick Staining Kit	Solarbio, Beijing, China	G15540
Hematoxylin Staining Solution	Solarbio, Beijing, China	G1080
0.45 μm Nitrocellulose Membrane	Solarbio, Beijing, China	Ya1711
DNase I	Solarbio, Beijing, China	D8070
DAPI	Solarbio, Beijing, China	C0065
RIPA Lysis Solution	Solarbio, Beijing, China	R0010
Collagenase I	Sigma-Aldrich, Chicago, IL, USA	C2674
Histopaque 1077	Sigma-Aldrich, Chicago, IL, USA	10771
Histopaque 1119	Sigma-Aldrich, Chicago, IL, USA	11191
Phorbol Myristate Acetate (PMA)	Sigma-Aldrich, Chicago, IL, USA	P8139
Fetal Bovine Serum	Gibco, Grand Island, NY, USA	10100147
RPMI 1640	Gibco, Grand Island, NY, USA	11835030
DMEM/F12	Hyclone Laboratories Inc., Logan, Utah, USA	Sh30023.Fs
RNAiso Plus	Takara, Tokyo, Japan	9109
BCA Protein Assay Kit	Takara, Tokyo, Japan	T9300A
Evo Reverse Transcription Kit	Accurate Biology, Changsha, China	AG11705
SYBR^®®^ Green Hs Master Mix qPCR	Accurate Biology, Changsha, China	AG11701
Mouse Anti-β-Actin Antibody	Abcam, Cambridge, UK	ab8226
Anti-Neutrophil Elastase Polyclonal	Abcam, Cambridge, UK	ab68672
Goat Anti-Rabbit IgG H&L (HRP)	Abcam, Cambridge, UK	ab97051
Alexa Fluor^®®^ 488 Anti-Rabbit H&L	Abcam, Cambridge, UK	ab150077
Mouse Anti-DNA-Histone1 Monoclonal	Millipore, Billerica MA, USA	Mab3864
Alexa Fluor 594 Anti-Mouse H&L	Proteintech, Chicago, IL, USA	SA00013-3
Rabbit Anti-Caspase-1 Antibody	Proteintech, Chicago, IL, USA	22915-1-Ap
Rabbit Anti-ASC Antibody	Proteintech, Chicago, IL, USA	10500-1-Ap
Rabbit Anti-NLRP3 Antibody	Proteintech, Chicago, IL, USA	19771-1-Ap
Rabbit Cytokeratin-18 Antibody	Proteintech, Chicago, IL, USA	10830-1-Ap
Rabbit Anti-Caspase-4 Antibody	Affinity Bioscience, Liyang, China	AF5130
Rabbit Anti-GSDMD Antibody	Affinity Bioscience, Liyang, China	Df12275
IRDye 800cw Rabbit IgG (H + L)	Gene Company Limited, HongKong, China	926-68070
IRDye 680rd Mouse IgG (H + L)	Gene Company Limited, HongKong, China	926-32211

## Data Availability

Due to data confidentiality requirements, the manuscript cannot provide raw data.

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
