# Peer review of "Neutrophil Extracellular Traps Mediate Bovine Endometrial Epithelial Cell Pyroptosis in Dairy Cows with Endometritis"

_ijms, 2022, doi:10.3390/ijms232214013_

Round 1

Reviewer 1 Report

Manuscript ijms-1932922

 Neutrophil Extracellular Traps Mediate Bovine Endometrial Epithelial Cells Pyroptosis in Dairy Cows with Endometritis

General

With an objective to investigate whether NETs cause pyroptosis of bovine endometrial epithelial cells during endometritis, the authors developed some consequences of contact NET-epithelial cells and measured pyroptosis-related proteins in endometrial tissue. The paper shows weaknesses because the authors took into consideration the only NET-dependent pyroptosis with a poor look into the pyroptosis pathway, including in the discussion.

Expression

The ms must be rewritten to follow grammatical rules, correct and appropriate spelling.

Some examples are given below.

Page 1, line 16; neutrophil induce. Same page, line 21; Using NETs 21 stimulate BEECs to validate. Same page line 36; it generally is considered

Page 2, line 50; components(including

Page 2, line 53; ‘dangerous signals’ is not appropriate if the authors consider a danger signal that refers to LPS, microorganisms, etc.

Page 2, line 55; and along the bodytext; DNase I is an enzyme with DNA as substrate, not an inhibitor of NETs

Page 2, line 67; characterized by cell swelling and an intact nucleus

Page 2, line 69; inflammasomes trigger cleavages depending on caspase-1

Page 6, Fig 1; page 7, Fig 3; page 8, Fig 4; page 9, Fig 5; please replace health by healthy control in the legends

Page 7, Fig 2, Fig 3; please do not present data in the legend, but as any figure must be interpreted by the reader without reference to the bodytext, please describe the experimental context (histological tissue, confocal or not, etc)

Page 8, line 305; As shown in the figure, please indicate number

Page 9, line 323; please use NETs

Page 10, Fig 6; please use NETs in the legends

Please correct the too abundant typos

Some texts of legends could be advantageously shortened; figures 5, 8

Comments

-1. Biological concept

-1.1. NETs do induce pyroptosis, but could not be limited as the only mediator of pyroptosis in endometritis development; other important inflammatory components, e.g. complement products, are displayed in the endometritis process. These are potent pyroptosis promoters/inducers and can also directly participate in the pathological process. Note that NETs may also deliver TLR activation signals, so innate immunity should be considered as such.

-1.2. Using DNAse I, an enzyme known to digest DNA component of NETs, has been developed as an only control in the experiments. The authors must introduce in the experimental model an inhibitor/inhibitors of pyroptosis or of pyroptosis pathway(s).

To identify NET-dependent pyroptosis is strategic for the paper. The authors must distinguish that is observed as molecular expression from that is corresponding to activation; because pyroptosis is presented as strategic for endometritis, they are recommended to consider activation and development of pyroptosis pathway in the experimental design.

-2. Abstract

Please display the interpretation of data after description of observations.

Please describe your observations and their interpretation appropriately; the reader must consider a demonstration where NETs is responsible of an increase of the level of pyroptosis-related proteins and proinflammatory cytokines, with related pathway(s), but not an only NETs formation along with pyroptosis-related proteins and some proinflammatory cytokines.

-3. Results

1.     Pages 6 and 7. The authors developed histopathological observations, subsection 3.2, and expression of some pro-inflammatory cytokines, subsection 3.3. They underlined the presence of neutrophils and plasma cells.

(i) How the authors identified and quantified both cell types is not clear;

(ii) Plasma cells have been selected in Fig 1 without reference to a particular involvement in pyroptosis, in endometritis or in an associated cytokine expression;

(iii) Because of the increased levels of some cytokines, the authors are invited to show the abundance of macrophages and T cells in biopsies, with identification of macrophage/T cell phenotypes to appropriately conclude the observation;

2.     Page 7, figure 2; page 10, figure 6. IFNs type-I, important inflammasome promoters and IFN-g, a prototypical proinflammatory cytokine, are missing. IL-17, important for neutrophil recruitment and promoting pyroptosis is also missing. Non-inflammatory cytokine or Th2 cytokine that could be used as control must be included.

A conclusion established on the results of the cytokine choreography with putative participation of T cell subsets is expected.

3.     Page 10, line 347. The title of subsection 3.7 refers to inflammation of epithelial cells. Because inflammation is a cell/tissue-integrated process, the authors are recommended to write a more appropriate title.

4.     Page 10, lines 355-369. The title of subsection 3.8 refers to cell pyroptosis through activation of some molecular intermediates. The authors used antibodies and displayed observations under a confocal laser microscope. If there evidence for an expression of some strategic molecules, the reader does not find any data in favour of their function.

Once more; please note that DNase-I is an enzyme, not an inhibitor.

5.     Page 10, line 370; page 13, lines 445-447. The authors conclude that NETs cause pyroptosis of epithelial cells; this is only cautiously suggested by some data using DNase-I, figures 7 and 8, in line with the use of conditional tense and careful language by the authors, page 10, lines 367-369. NETs are presented as to mediate BEECs pyroptosis ; participation of NETs in pyroptosis should be more appropriate.

6.     Page 13, lines 451-452. The referee did not find in vivo evidence that restraining NETs formation decreases endometrial tissue damage during endometritis. Please discard.

-4. Discussion

The authors are invited to develop a discussion, particularly for the experimental design for pyroptosis induction, in a comparison with recent data from Feng M et al. Int J Mol Sci 2022 ; 23 : 10124. doi : 10.3390/ijms231710124.

The authors are recommended to introduce and discuss the involvement of complement that has been shown to directly activate the NLRP3 component of inflammasome.

-5. Conclusion

Pyroptosis is a form of programmed cell death that may account for endometritis. The ms is dealing with pyroptosis induction by NETs, with relevance to endometritis. This issue could be attractive by many readers.

Pertaining to this objective, the article reports data from experiments limited to the only contact of NETs to endometrial epithelial cells. As NETs are a heterogeneous material and could be associated in vivo with products known as pyroptosis inducers, incl. complement products, the outcomes of the results are limited.

Important: There are frequent inadequacies in the expression that must be corrected to display an appropriate wording. So the present form of the article does not meet the requirements of the journal.

Author Response

Dear  Reviewers:

Thank you for your letter and for the reviewers’ comments concerning our manuscript entitled “Neutrophil Extracellular Traps Mediate Bovine Endometrial Epithelial Cells Pyroptosis in Dairy Cows with Endometritis” (ID: ijms-1932922). Those comments are all valuable and very helpful for revising and improving our paper, as well as the important guiding significance to our researches. We have studied comments carefully and have made correction which we hope meet with approval. Revised portion are marked in red in the paper. The main corrections in the paper and the responds to the reviewer’s comments are listed in the word file of Author's Reply to the Review Report (Reviewer 1).

Reviewer 2 Report

In the manuscript entitle “Neutrophil Extracellular Traps Mediate Bovine Endometrial Epithelial Cells (BEEC) Pyroptosis in Dairy Cows with Endometritis” aimed to analyze the induction of Pyroptosis in BEEC by NETs. For that the authors analyzed the endometrial tissue of cows with endometritis and NETs induced cytotoxicity in BEEC. The authors show an adequate amount of data, the manuscript is well written, and the statistical analysis is well performed. However, the manuscript needs to be improved in some issues as mentioned below.

 1.       Authors showed an increment of Proinflamatory cytokines and different proteins related to NETs and pyroptosis either by ELISA and Neutrophils infiltration by Immunohitoquemistry. They also indicated an increment of NETs, Neutrophil Elastase and Histone-1 by immunofluorescence. theses parameters need to be quantified and statistics should be displayed (figures 3 and 7). In addition, the authors need to indicate in the methods how the quantification of these parameters is carried out, both in immunohistochemistry and immunofluorescence.

2.       The authors indicated that NETs induced pyroptosis in BEECs, for that authors showed and increment on Caspase-1 and Caspase-4 either in tissues or BEEC. The authors showed the quantification of caspases, but they need to show and increment of cleaved Caspases, either for capase-1 and capase-4, by western blots to indicate and demonstrate and increment of caspases activation. These would not need new experiments but the analyses of the full length or cleaved caspase from the cells and tissues extracts

3. The authors analyzed the induction of pyroptosis in BEEC by NETs, they showed incremented BEEC cytotoxicity and inflammatory cytokines, GSDMD-N, etc., in BEEC treated with NETs, thew also showed that the Treatment with DNAase-1 reduced the amount of this protein in BEEC. These analyses were performed by Western blots. They also showed the reduction of NETs and some of these proteins by immunofluorescence, as previously mentioned quantification of these data needs to be performed (figure 7).

4.       The authors analyzed the induction of pyroptosis in BEEC by NETs, they showed incremented BEEC cytotoxicity and inflammatory, GSDMD-N, etc in BEEC treated with NETs, and treatment with DNAase-1 reduced the amount of this protein either by Western blot, and in figure 7 they show the reduction of NETs and some of these proteins by immunofluorescence as previously mentioned the quantification of these data needs to be performed.

5.       The supplementary figures were not provided.

6.       Line 307, there is mistakes in the text “that in cows 307 in the healthy group (Figures 4 and 5)”.

Author Response

Dear  Reviewers:

Thank you for your letter and for the reviewers’ comments concerning our manuscript entitled “Neutrophil Extracellular Traps Mediate Bovine Endometrial Epithelial Cells Pyroptosis in Dairy Cows with Endometritis” (ID: ijms-1932922). Those comments are all valuable and very helpful for revising and improving our paper, as well as the important guiding significance to our researches. We have studied comments carefully and have made correction which we hope meet with approval. Revised portion are marked in red in the paper. The main corrections in the paper and the responds to the reviewer’s comments are listed in the word file of Author's Reply to the Review Report (Reviewer 2).

Round 2

Reviewer 1 Report

Manuscript ijms-1932922

Neutrophil Extracellular Traps Mediate Bovine Endometrial Epithelial Cells Pyroptosis in Dairy Cows with Endometritis

The authors answered the questions and introduced some amendments in the manuscript.

However shortcomings persisted.

-1. Neutrophils, plasma cells.

-1.1. How did the authors estimate an abundance of neutrophils in endometritis tissues?

-1.2. Plasma cells are presented as one of the hallmarks of endometritis and not present in a healthy endometrial tissue; the authors advantageously indicated in Fig 1 some cells with deeply stained nuclei, wheel-mounted, with less cytoplasm. As other nucleated cells also appear as such, the authors are recommended to unambiguously document this observation; e.g. absence of CD19 and CD20, high expression of CD139 and CD319.

-2. Macrophages and T cells in biopsies. The authors are invited to describe the observations including the limits of the methods they adopted ; e.g. without identification of a macrophage and T cell phenotype.

-3. Accumulation of neutrophils. Neutrophils are the main source of NETs and the authors did not investigate Th17. So where are neutrophils coming from?

Expression

Legend of Fig 6. Please write cytokine production

Subsection 2.8, line 390. Please write participate in BEEC pyroptosis

Section 3, line 440. Please write Thus, we

The authors are invited to take an additional look to correct typos and errors.

Author Response

Dear reviewer :

On behalf of my co-authors, I am very grateful to the editors and reviewers for their positive and constructive comments and suggestions on our manuscript entitled "Neutrophil Extracellular Traps Mediate Bovine Endometrial Epithelial Cells Pyroptosis in Dairy Cows with Endometritis”. (ID: ijms-1932922).

We have carefully studied the reviewers' comments and have revised the discussion section and results section of the manuscript accordingly to the suggestions given by the academic editors and reviewers, and have marked the manuscript changes with traceable marks in the paper. We have tried our best to revise our manuscript according to the comments. The main corrections in the paper and the responses to the reviewer’s comments are as flowing: Author's Reply to the Review Report(please see in the attachment).

We are very grateful to you and the reviewers for your comments on our paper. We look forward to hearing from you!

Reviewer 2 Report

The authors responded to most of the issues raised, and the manuscript is much improved. I recommend its publication, although some issues need to be corrected. The new version of the manuscript is difficult to follow in the presented format, as the text is not clearly visible. There are some incorrect wordings in the abstract and in the text that need to be improved. Regarding the answer to question 2, the authors indicated that the activated form is detected, but figure 5 and 8 and the original western-blot, non-published data, only the full-lengh caspases (45kD) can be observed, but did not observe the digested caspases (10, kD, 20Kd, or 30Kd). The authors correctly indicated in the text that there is an increase in the caspase 1 and 4 by Western blot.

I suggest indicating increment of the expression of these caspases in the immunofluorescence data as well. Regarding to figure 7 text and figure 7  legend, and in the results section 2.8 , “2.8. NETs induced BEEC pyroptosis through the activation of NLRP3, ASC, caspase-1, and 375 caspase-4“, the activation term is indicated. If so, the authors need to indicate either in the figure legend and in the material and methods the clone used to detect these proteins. This need to be indicated also for all the antibodies used. On the other hand, it would be useful to indicate why increased activation of each of the proteins is observed in the text and in the material and methods.

Author Response

Dear reviewer:

On behalf of my co-authors, I am very grateful to the editors and reviewers for their positive and constructive comments and suggestions on our manuscript entitled "“Neutrophil Extracellular Traps Mediate Bovine Endometrial Epithelial Cells Pyroptosis in Dairy Cows with Endometritis”. (ID: ijms-1932922).

We have carefully studied the reviewers' comments and have revised the discussion section and results section of the manuscript accordingly to the suggestions given by the academic editors and reviewers, and have marked the manuscript changes with traceable marks in the paper. We have tried our best to revise our manuscript according to the comments. The main corrections in the paper and the responds to the reviewer’s comments are as flowing: Author's Reply to the Review Report(please see in the attachment).

We are very grateful to you and the reviewers for your comments on our paper. We look forward to hearing from you!

Round 3

Reviewer 1 Report

The authors answered the comments and introduced appropriate amendments in the manuscript.

I agree to the publication of the last version of the manuscript.